# Xenopus Sox11 Partner Proteins and Functional Domains in Neurogenesis

**DOI:** 10.3390/genes15020243

**Published:** 2024-02-15

**Authors:** Kaela S. Singleton, Pablo Silva-Rodriguez, Doreen D. Cunningham, Elena M. Silva

**Affiliations:** 1Interdisciplinary Program in Neuroscience, Georgetown University Medical Center, Washington, DC 200057, USA; 2Department of Biology, Georgetown University, Washington, DC 20057, USA; ps977@georgetown.edu (P.S.-R.); ddc24@georgetown.edu (D.D.C.)

**Keywords:** cell differentiation, neurogenesis, *Xenopus laevis*, transcription factor, protein interaction

## Abstract

Sox11, a member of the SoxC family of transcription factors, has distinct functions at different times in neural development. Studies in mouse, frog, chick, and zebrafish show that Sox11 promotes neural fate, neural differentiation, and neuron maturation in the central nervous system. These diverse roles are controlled in part by spatial and temporal-specific protein interactions. However, the partner proteins and Sox11-interaction domains underlying these diverse functions are not well defined. Here, we identify partner proteins and the domains of *Xenopus laevis* Sox11 required for protein interaction and function during neurogenesis. Our data show that Sox11 co-localizes and interacts with Pou3f2 and Neurog2 in the anterior neural plate and in early neurons, respectively. We also demonstrate that Sox11 does not interact with Neurog1, a high-affinity partner of Sox11 in the mouse cortex, suggesting that Sox11 has species-specific partner proteins. Additionally, we determined that the N-terminus including the HMG domain of Sox11 is necessary for interaction with Pou3f2 and Neurog2, and we established a novel role for the N-terminal 46 amino acids in the specification of placodal progenitors. This is the first identification of partner proteins for Sox11 and of domains required for partner-protein interactions and distinct roles in neurogenesis.

## 1. Introduction

The Sry-related HMG box (Sox) family of transcription factors plays critical regulatory roles in the development of metazoans [1,2]. All Sox proteins contain a high mobility group (HMG) domain that binds both partner proteins and DNA in order to alter expression of downstream target genes [3,4]. Sox transcription factors are grouped into eight subfamilies (A–H) based on sequence homology and functional similarity. The SoxB and SoxC subfamily proteins play essential roles in orchestrating neurogenesis in the central nervous system (CNS). SoxB1 proteins drive neural progenitor specification and maintenance, and SoxC proteins promote differentiation and neuron maturation [5,6,7]. Both SoxB and SoxC proteins have multiple and seemingly opposite roles during neurogenesis. For example, the SoxB2 protein Sox21 expands the neural progenitor population when overexpressed but is also required for differentiation [8]. Similarly, the SoxB1 protein Sox2 is implicated in not only embryonic stem cell specification and maintenance but also neuron maintenance [9,10,11]. Thus, functional studies in numerous organisms demonstrate that Sox proteins perform unique functions in different neural cell types and at various times during neurogenesis.

Across subfamilies, the HMG domain of Sox proteins is more than 50% identical, resulting in all Sox proteins binding a similar DNA motif with low affinity [12]. Thus, interactions with partner proteins are required to facilitate specific and high-affinity binding of Sox proteins to DNA regulatory regions. The most commonly identified partners of Sox proteins in neurogenesis are Pou and E-Box proteins [13,14,15]. For example, Sox2 complexes with Pou5f1, Pou3f2, and Neurog2 at various stages in development and drives the expression of different genes in multiple developmental processes [8,16,17,18]. Sox2 cooperates with Pou5f1 (Oct3/4) to control embryonic stem cell differentiation and later complexes with Pou3f2 (Oct7) to promote neural specification [19,20,21]. Sox proteins can also form homo- and hetero-dimers; Sox2 interacts with Sox21 to promote ectodermal cell fate in stem cells [17]. Additionally, Sox2 and Sox21 each bind Neurog2 to promote or inhibit neural differentiation, respectively [8,22]. These data indicate that identifying and characterizing partner-protein interactions is essential to understanding the regulation of Sox protein function.

Sox proteins use various domains to complex with partner proteins. While many transcriptional partner proteins interact with the Sox HMG domain [15,20,23,24,25,26], some interact with multiple domains or a domain outside of the HMG. Sall4 interacts with the C-terminus, N-terminus, and HMG domain of Sox2 in embryonic stem cells. Others, like HDAC1, interact with only the C-terminus of Sox2 [27]. It has been proposed that protein interactions with domains outside of the HMG serve to stabilize binding of other transcriptional partner proteins [15,28]. The reliance on partner proteins for specificity allows two levels of regulation of Sox protein function: availability of partner proteins within a given cell or tissue type and the relative affinity of partner-protein interactions. Through identification of both Sox partner proteins and interaction domains, we can understand how Sox proteins precisely control gene networks during neurogenesis. 

In this study, we identify partner proteins as well as the interaction and functional domains of the SoxC protein Sox11 during neurogenesis. Work from our lab and others has shown that Sox11 is involved in neural induction and in neuron differentiation and maturation [29,30,31,32]. We recently characterized Sox11 expression and function in the *Xenopus laevis (Xl)* neural plate and mouse cortex [33]. *Sox11* expression is found throughout the developing neural plate and cortex and promotes neuronal maturation in both mouse cortical development and *Xl* neurogenesis. Surprisingly, we discovered that Sox11 is not functionally interchangeable between these two species, in contrast to many other Sox transcription factors [33,34]. This suggests that Sox11 is essential for neuron formation across species, but the molecular mechanism underlying Sox11 function is not conserved. 

To investigate this and to better understand how *Xl* Sox11 drives neurogenesis, we establish that *Xl* Sox11 interacts with Neurog2 and Pou3f2/Brn2 but not Neurog1, all known partners of Sox11 in mouse cortex [30]. We also identify the domains of Sox11 needed for both partner-protein binding and formation of neural progenitors and neurons. Collectively, our data show that the first 46 amino acids of Sox11 are required for a strong interaction with Neurog2 and Pou3f2 and formation of the posterior placodes. Furthermore, the HMG domain of Sox11 is required for both partner-protein binding and Sox11 function. Additionally, the C-terminus and transcription activation domain of Sox11 are necessary for Sox11 to induce the formation of neurons in the developing neural plate. These data suggest that Sox11 engages species-specific partner proteins to drive neuron formation, and in *Xenopus*, it interacts with different partners to promote various functions in different regions of the developing neural plate. This study represents the first identification of partner proteins for *Xl* Sox11 and the first characterization of Sox11 protein interaction domains and their contribution to neurogenesis.

## 2. Materials and Methods

### 2.1. Plasmids

Mouse NEUROG2-MYC (gift from Qiang Lu), Pou3f2-HA, Neurog1-HA (generated by GeneWiz, Burlington, MA, USA), Neurog2-MYC (gift from Sally Moody), mouse Sox11-FLAG (gift from Maria Donoghue), and Sox11.S-FLAG were used. Sox11-FLAG mutants were created by site-directed mutagenesis (Agilent, Santa Clara, CA, USA). Each plasmid expressed the correct-sized protein as determined by immunoprecipitation (IP) and western blot (WB) analysis.

### 2.2. Identifying Cells Using Xenopus Time Series Database

Single-cell transcriptome measurements were taken from the *Xenopus* Jamboree database (kleintools.hms.harvard.edu/tools/spring.html, accessed on 2 May 2020). The threshold for *sox11*, *neurog*, and/or *pou3f2* expression was set to 0.50 to include cells expressing low levels of all three genes.

### 2.3. Xenopus Animal Usage and Embryo Manipulation

All frog use and care were in accordance with federal and institutional guidelines. Frog embryos were obtained using standard methods [35,36] and staged according to Nieuwkoop & Faber. 

### 2.4. Whole-Mount In Situ Hybridization

Whole-mount in situ hybridization was performed as previously described [37,38].

### 2.5. Frog Microinjection

mRNAs used for injection were made in vitro using the mMESSAGE mMACHINE^®^ Transcription Kit (Life Technologies, Carlsbad, CA, USA). Embryos were injected with mRNA (1.2 ng) and 1:5 Dextran (tracer, ThermoFisher, Waltham, MA, USA) in one cell of two-cell stage embryos to over-express *sox11* or deletion constructs. Embryos were cultured until stage 14 in 1/3 MMR at 18 °C and fixed in Bouin’s fixative. Embryos were visualized under 488 nm wavelength fluorescence to identify the injected side.

### 2.6. In Vitro Translation (IVT) and Co-Immunoprecipitation (co-IP)

The TNT^®^ SP6 High-Yield Wheat Germ Protein Expression System (Promega, Madison, WI, USA) was used to confirm translation of mRNA and determine relative protein quantity. Protein products were denatured at 65 °C for 10 min and separated on an SDS-PAGE gel. Antibodies were anti-FLAG-HRP (Sigma-Aldrich, St. Louis, MO, USA, A8592, mouse monoclonal 1:2500), anti-HA-HRP (Roche, Basel, Switzerland, 12013819001, rat monoclonal 1:2500), and anti-cMyc-HRP (Abcam, Waltham, MA, USA, ab19312, rabbit polyclonal 1:2500).

For co-IP, IVT reactions were carried out with 500 ng of mRNA; 66% of the IVT reaction was subjected to co-IP, and 33% was saved as input; co-IP was performed as previously described [8] using 2 μg/mL of anti-FLAG (Sigma A8592, mouse monoclonal), anti-HA (Cell Signaling, Danvers, MA, USA, C29F4, rabbit monoclonal), or anti-MYC (Cell Signaling, D84C12, rabbit monoclonal). Samples were separated via SDS-PAGE on 12% pre-cast gels (Bio-rad, Hercules, CA, USA).

### 2.7. Western Blotting

Western blotting was performed using the Bio-Rad Mini TransBlot Transfer System, with the Bio-Rad PVDF Transfer Kit. Blots were incubated with anti-FLAG-HRP, anti-HA-HRP, anti-myc-HRP, and anti-β actin (Sigma, A2228, mouse monoclonal, 1:5000) primary antibodies and Pierce ECL Plus chemiluminescent substrate (ThermoFisher), imaged using the ImaqeQuant LAS-4000 mini digital imager (GE Healthcare, Salt Lake City, UT, USA), and visualized for a maximum of 5 min. Input blots were run as a control for each experiment. Relative band density co-immunoprecipitation was determined as previously described [39,40].

## 3. Results and Discussion

Our previous overexpression studies demonstrated that even though *Xl* Sox11 and mouse SOX11 (mSOX11) promote neural differentiation in their respective species, neither promotes differentiation in the other species [33]. To determine if *Xl* Sox11 and mSOX11 utilize different partner proteins and, therefore, different mechanisms to promote differentiation in the *Xenopus* neural plate and mouse cortex, we asked if *Xl* Sox11 interacts with three proteins that bind to mSOX11 in the mouse cortex: Neurog1, Neurog2, and Pou3f2/Brn2 [30].

### 3.1. Sox11, Neurog, and Pou3f2 Are Co-Expressed in Distinct Cell Types of the Neural Plate

Whole mount in situ hybridization (WISH) results for *sox11*, *pou3f2*, *neurog1*, and *neurog2* have been previously published and are available from the online *Xenopus* database Xenbase [41]. To illustrate the overlapping expression of *pou3f2*, *neurog2*, and *sox11* in the neural plate and developing neural tube, we show expression in late gastrula and neurula embryos [41] (Figure 1A). *Sox11* is expressed throughout the neural plate, whereas the proneural protein genes *neurog1* and *neurog2* are both expressed in three stripes of cells fated to become the motor, inter, and sensory neurons from medial to lateral. Notably, *pou3f2* expression is undetectable at stage 12 by WISH due to low transcript levels, but it is visible at stage 14 in the anterior neural plate [42].

An interesting observation from single-cell transcriptome measurements in the open access database Jamboree [43] is that the majority of *neurog+* and *pou3f2+* cells also express *sox11* in neurula stage 13/14 embryos (Figure 1B). However, due to a low sequencing depth, we could not distinguish between *neurog1*, *neurog2*, or *neurog3*, as *neurog*^+^ cells are grouped together. Most striking is in early neurons (EN), where 90% of *neurog+* cells (413/458 cells) are *sox11*+, and in the neural plate, 88% of *pou3f2+* cells are *sox11+* (141/160 cells; Figure 1B). However, very few cells (<25) are positive for all three markers: *sox11*, *neurog*, and *pou3f2*. These findings suggest that *sox11* is co-expressed in distinct cell types with *neurog* and *pou3f2*. Whereas the majority of early neurons co-express *sox11* and *neurog* (413/585), the majority of the *pou3f2+*/*sox11+* cells are located in the anterior neural plate, where there are very few *sox11+*/*neurog+* cells. These single-cell data suggest that there are two populations of *sox11*-expressing cells: one that co-expresses *neurog* and another that is *pou3f2+*.

### 3.2. Sox11 Partners with Neurog2 and Pou3f2 but Not Neurog1

To examine the interactions of Sox11 with potential partner proteins, we conducted co-immunoprecipitation experiments using in vitro translated, epitope-tagged proteins: Sox11-FLAG, Pou3f2-HA, Neurog1-HA, and Neurog2-MYC. Our results indicate that Sox11-FLAG interacts with both Pou3f2-HA and Neurog2-MYC (Figure 2A,B) but not with Neurog1-HA (Figure 2C).

In summary, *Xl* Sox11 interacts with Neurog2 and Pou3f2 but not with Neurog1. This differs from mouse SOX11, which interacts with all three proteins. Even though *Xl* Sox11 and Neurog1 did not interact in our in vitro system, they may require post-translational modification (PTM) or a co-factor to interact. There is precedent for PTMs of mouse Sox11 in developing mouse retina and hippocampus [44,45], although to our knowledge, no studies have investigated PTMs of Sox11 or Neurog1 during neurogenesis. Furthermore, Neurog1 complexes with CBP/p300, which facilitates an interaction with Smad1 to inhibit glial cell differentiation and promote neurogenesis [46]. These modifications and interactions are absent in the in vitro system. As antibodies for *Xenopus* proteins become available, Sox11 protein interactions can be tested at different stages in the developing frog embryo.

These data support the idea that Sox11 has a different array of partner proteins in the mouse and frog during neurogenesis, especially considering that overexpression of mouse SOX11 has no effect on frog embryos and therefore does not recapitulate the overexpression phenotype of *Xl* Sox11 [33]. Since the DNA-binding domains of Sox proteins are highly homologous across species (100% amino acid identity of Sox11 DNA-binding domains in mammals, chicks, and zebrafish), it was proposed that the differential function of mouse and frog Sox11 was due to a single amino acid change from K to N in the frog DNA-binding domain at position 91 of the protein and position 43 of the HMG DNA-binding domain [33]. Based on currently available sequences, we have found that this amino acid difference is unique to anura amphibians (frogs). However, it is also worth noting that mouse and frog Neurogenins share only 36.3% similarity with eight amino acids divergent in the 57aa DNA-binding domain. Domain swap experiments will be used to determine if the amino acid changes in the DNA-binding domains lead to differences in protein interaction and function.

### 3.3. Sox11 N-Terminus and HMG Domain Are Necessary for Protein–Protein Interactions

To investigate which domains of Sox11 are necessary for partner-protein binding, we generated three deletion constructs that express modified Sox11-FLAG protein: ΔN46-Sox11-FLAG (lacking the first 46 amino acids upstream of the HMG domain), ΔHMG-Sox11-FLAG (lacking the 72 amino acid HMG domain), and ΔCterm-Sox11-FLAG (containing only the N-terminus and HMG of Sox11 after removal of 265 C-terminal amino acids) (Figure 3A). We performed co-IP analysis of these modified Sox11 proteins with Neurog2 and Pou3f2, using FLAG, HA, or MYC antibodies, and examined them through Western blotting.

Interestingly, the absence of the Sox11 C-terminus does not impact binding with Pou3f2-HA or Neurog2-MYC (Figure 3B,E). This suggests that the C-terminus is not necessary for the interaction of Sox11 with these two proteins. In contrast, the removal of the N46 domain decreases the interaction with both partner proteins (Figure 3C,F), and loss of the HMG domain results in no detectable interaction with these proteins (Figure 3D,G). We further normalized co-expression bands to the input of three replicates and quantified the relative band density [39,40]. The relative level of proteins of three replicates are graphically represented in Figure 3H,I. The graph reveals a reduction in the amount of protein recovered with immunoprecipitation of Neurgo2-MYC or Pou3f2-HA with ΔNterm-Sox11-FLAG or ΔHMG-Sox11-FLAG (Figure 3H,I, green and orange bars, respectively). These findings indicate that the 46 amino acid N-terminus of Sox11 is required for strong partner-protein interactions with Pou3f2 and Neurog2, while the HMG domain is essential for partner-protein binding.

The HMG domain’s significance for protein interaction is not unique to Sox11 but extends to other Sox proteins. For instance, SoxE proteins (Sox8, Sox9, and Sox10) require the C-terminal tail of the HMG domain to complex with partners [23,47]. However, in other cases, domains outside of the HMG are required for partner-protein interactions. For example, the B-homology domain and C-terminus of Sox2 are required for partner interactions with stem cells [27], and Sox18 binds to MEF2C in endothelial cells through its C-terminal domain [48]. Thus, similar to other Sox proteins, Sox11 utilizes domains beyond the HMG to complex with partners, likely contributing to its unique binding specificity [15].

### 3.4. Sox11 C-Terminus Is Required for Neuron Formation

Our prior research demonstrated that Sox11 plays a critical role in primary neurogenesis, with overexpression increasing both neural progenitors and neurons and MO knockdown decreasing neurons [33]. Together, these data establish Sox11 as a critical protein during early neurogenesis.

To determine the functional significance of the Sox11 protein domains, we analyzed the effect of the Sox11 deletion proteins on neurogenesis (Figure 4). For this analysis, we injected each mRNA into one cell of a two-cell blastomere embryo and examined changes in neurula embryos using *WISH* for *tubb2b*, a marker of neurons, and for *sox3*, a marker of neural progenitors. If a domain is necessary for Sox11 function, misexpression will not alter *tubb2b* expression. Conversely, the loss of a domain that is not essential for Sox11 function would drive an increase in *tubb2* and/or *sox3* consistent with our previous gain-of-function findings [33].

Our data reveal that overexpression of *Δcterm-Sox11* does not impact *sox3* expression but slightly reduces *tubb3* expression. This suggests that ΔCterm-Sox11 functions as a dominant negative protein and, like the morpholino knockdown, reduces neuron formation [33]. On the other hand, overexpression of *Δn46-Sox11* leads to ectopic expression of *tubb2b* in the neural plate resembling the effect of full-length *Sox11*, albeit to a lesser extent. In addition, *sox3* expression is lost in the placodes (Figure 4, asterisk). These results indicate that the 46 N-terminal amino acids are not only required for full activity in the neural plate but also crucial for placode development. It is possible that ΔN46-Sox11 functions as a dominant negative in the placode, which necessitates these 46 aa for proper placode development. One hypothesis is that ΔN46-Sox11 fails to effectively bind a partner protein essential for placode formation and instead interferes with the function of endogenous Sox11.

Next, we explored the function of two internal domains—the HMG DNA-binding domain and the serine-rich transactivation domain (TAD) in the C-terminus. Numerous studies have shown that the HMG domain of many Sox proteins is essential to both DNA- and partner-protein binding [49,50]. Therefore, as expected, overexpression of *Δhmg-Sox11* did not mimic full-length Sox11 and increase expression of either *tubb2b* or *sox3* but did lead to a decrease in expression of *tubb2b* in the trigeminal ganglion in 15/27 embryos (asterisk). Additionally, we explored the role of the 48 amino acid TAD [13,51,52]. We found that in the absence of the TAD, overexpression of Sox11 decreased *tubb2b* in the trigeminal ganglia in 56% of the embryos (asterisk) but did not affect *sox3* expression or *tubb2b* expression in the neural plate. This aligns with prior studies showing TAD’s essential role in Sox11 function in vitro. Sox11 is known as the most potent transactivator of the SoxC family and has even been shown to be more potent than Sox2 [52].

Our functional analysis of Sox11 domains has uncovered embryo phenotypes that support the involvement of protein domains in protein interactions outside of the HMG DNA-binding domain. Overexpression of ΔCterm-Sox11 replicates the knockdown phenotype, resulting in reduced neuron formation (Figure 4). This effect could be due to the sequestration of partner proteins in the neural plate, hindering endogenous Sox11 from interacting with these proteins. Since ΔCterm-Sox11 lacks the TAD, these complexes are unable to activate target genes.

Additionally, we show that excess ΔN46-Sox11 functions similarly to Sox11 overexpression in embryos, leading to increased neurons, albeit less effectively. This aligns with the weak interaction of ΔN46-Sox11 and Neurog2. We also show that overexpression of ΔN46-Sox11 decreases neural progenitors as marked by *sox3* in the placodes (Figure 4, arrow). To further investigate this result, we used the single-cell sequencing database [43]. We identified posterior placodal cells as those expressing *pax8*, *six1*, and *sox9* and found that 487 out of 618 placodal cells co-expressed *sox11*, supporting the role for Sox11 in placodal development as previously described [53,54]. We confirmed that neither *pou3f2* nor *neurog* is expressed in the posterior placode. Thus, Sox11 likely interacts with an unknown placodal partner protein to regulate posterior placodal development. These findings underscore the essential role of the C-terminus of Sox11 in neuron formation, emphasizing the need for further research to elucidate Sox11 partner proteins in placodal progenitors.

In total, these functional deletion studies represent the first in vivo characterization of Sox11 domains to determine their contribution to neurogenesis. Together, our data suggest that the HMG domain is necessary for Sox11 function during neuron formation, the N46 domain is necessary for development of neural progenitors in the placodes, and the C-terminus of Sox11, including the TAD, is necessary for neuron formation.

## 4. Conclusions

Sox transcription factors cooperate with region-specific partner proteins to regulate downstream targets and orchestrate neurogenesis. To identify and characterize Sox11 partner-protein interactions essential to neurogenesis, we tested whether Sox11 partner proteins are conserved between mouse and *Xenopus* and identified the domains of Sox11 necessary for protein interaction and function. In conclusion, here, we uncover several critical features of Sox11, a protein necessary for neurogenesis. First, Sox11 is co-expressed with Pou3f2 and Neurog2 in the anterior neural plate and early neurons, respectively. Second, Sox11 partner proteins are not conserved across species leading to the enticing possibility that changes in SoxC proteins evolved to enable expansion of the cortex in mammals. Third, the HMG domain and the first 46 amino acids in the N-terminus are necessary for strong partner-protein interactions, whereas the C-terminus plays no role in the binding of Pou3f2 and Neurog2. Lastly, we show that the C-terminus of Sox11, and specifically, the TAD, is required for promoting neuron formation and that the N46 domain of Sox11 is essential in posterior placodal development. To our knowledge, these are the first partner proteins identified for *Xenopus* Sox11 and the first identification of Sox11 domains essential for protein interaction and neuron formation in the developing neural plate.

## Figures and Tables

**Figure 1 genes-15-00243-f001:**
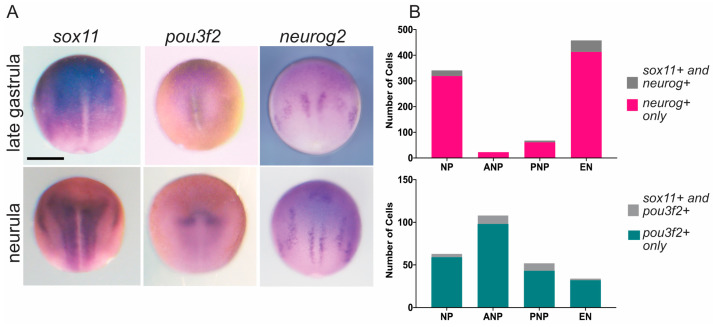
*Sox11*, *neurog2*, and *pou3f2* are co-expressed in distinct cell types of the neural plate. (**A**) WISH of *sox11*, *pou3f2*, and *neurog2* at stage 12.5 (late gastrula) and stage 14 (early neurula). Embryos are dorsal view with anterior to the top. *Sox11* is expressed throughout the neural plate and later in placodes. *Pou3f2* expression is not detectable until the neurula stage and is strongest in the anterior neural plate. *Neurog2* is expressed in the stripes of the neuronal progenitors. The scale bar is 500 microns. (**B**) Total number of cells across developmental stages that are *sox11*^−^ and *neurog+* (dark gray), *sox11+* and *neurog+* (magenta) (top) or *sox11*^−^ and *pou3f2+* (light gray), and *sox11+* and *pou3f2+* (teal) (bottom) within the neural plate (NP) at stage 12 and within the anterior neural plate (ANP), posterior neural plate (PNP), and early neurons (EN) at stage 13/14.

**Figure 2 genes-15-00243-f002:**
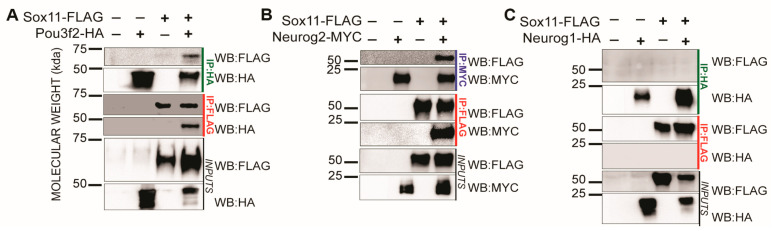
*Xl* Sox11 interacts with Pou3f2 and Neurog2, but not Neurog1. (**A**–**C**). Immunoprecipitation (IP) of *Xl* Sox11-FLAG and Pou3f2-HA (**A**), Neurog2-MYC (**B**), or Neurog1-HA (**C**) from in vitro translated proteins. Proteins were immunoprecipitated using either FLAG (red), HA (green), or MYC (blue) antibodies. Samples were analyzed by western blotting (WB) indicated on the right with FLAG-HRP, MYC-HRP, or HA-HRP. Input represents the total protein lysate.

**Figure 3 genes-15-00243-f003:**
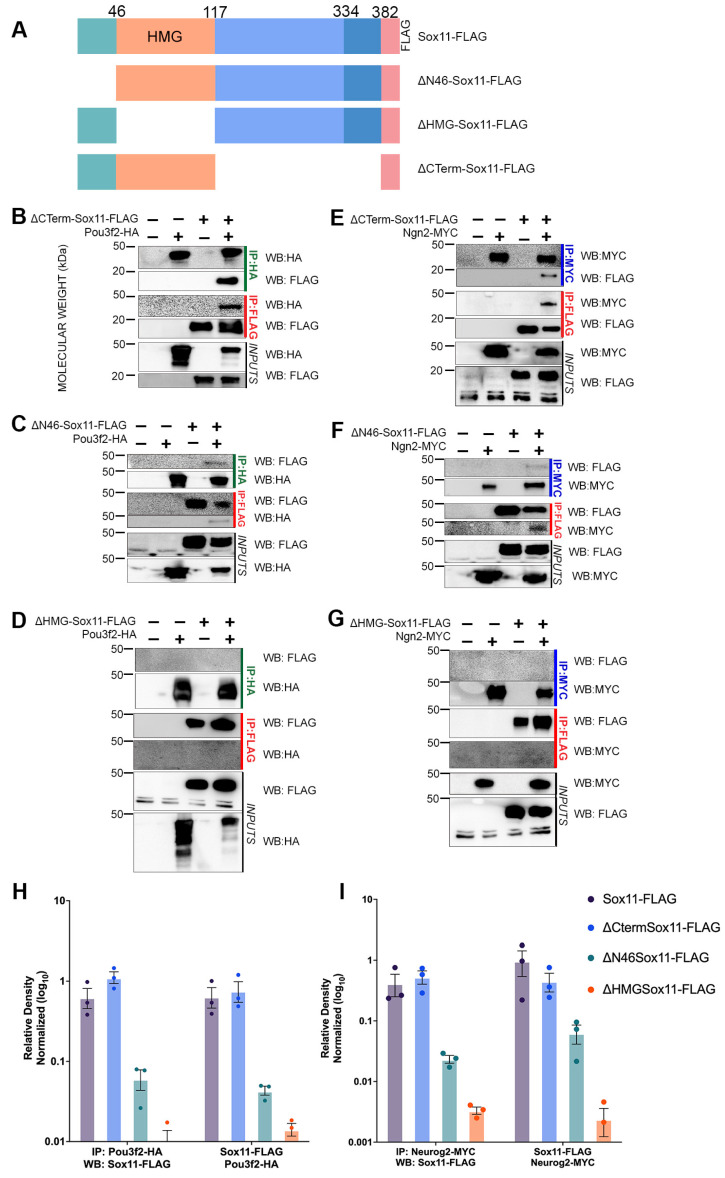
Sox11 N-terminus is essential for protein–protein interactions. (**A**) Schematic of Sox11 and deletion constructs with referenced domains marked. ΔN46-Sox11-FLAG lacks the 46 amino acids (teal) upstream of the HMG domain. ΔHMG-Sox11-FLAG lacks the 72 amino acid HMG domain (orange), and ΔCterm-Sox11-FLAG lacks 265 amino acids (blue) and consists of the N-terminus and HMG domain. (**C**–**G**) Immunoprecipitation (IP) of ΔCterm-Sox11-FLAG, ΔN46-Sox11-FLAG, or ΔHMG-Sox11-FLAG with Pou3f2-HA or Neurog2-MYC (Ngn2-MYC). Proteins were generated via in vitro translation and immunoprecipitated using either FLAG (red), HA (green), or MYC (blue) antibodies. Samples were analyzed by WB with anti-FLAG-HRP, anti-HA-HRP, or anti-MYC-HRP. Inputs demonstrate each protein in the extract. (**H**) Graphical representation of protein co-immunoprecipitated (Sox11 variants and Pou3f2) from three replicates, one of which is represented in (**B**–**D**). (**I**) Graphical representation of protein co-immunoprecipitated (Sox11 variants and Neurog2) from three replicates, one of which is represented in (**E**–**G**). Co-expression bands were normalized to the input in each sample. Data represent the mean ± sem on a log_10_ scale with percent of pulldown across 3 experimental replicates.

**Figure 4 genes-15-00243-f004:**
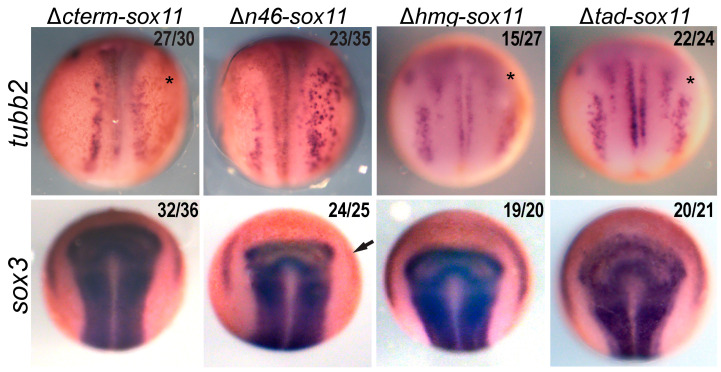
Identification of Sox11 domains required for placode and neuron formation. WISH of neurula (stage 14/15) embryos injected in one of two cells (dorsal view, anterior to the top) with Dextran as a tracer and either *Δcterm-Sox11*, *Δn46-Sox11*, *Δhmg-Sox11*, or *Δtad-Sox11* mRNA. The right side is the injected side, and the left side serves as control expression. Embryos were analyzed for expression of *tubb2* for neurons or *sox3* for neural progenitors. An arrow marks the reduction in *sox3* expression in placodal progenitors, and an asterisk marks the loss of expression in the trigeminal placode. Numbers in the upper right of each image denote the number of embryos with the phenotype over the total analyzed. A one-sample proportion test reveals that the null hypothesis that the overexpression of the mRNA has no effect on *tubb2* expression can be rejected in all cases except for the loss of the trigeminal placode for *Δhmg-Sox11 p*-value = 0.564.

## Data Availability

The data are contained within the article.

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
