# Peer review of "Xenopus Sox11 Partner Proteins and Functional Domains in Neurogenesis"

_genes, 2024, doi:10.3390/genes15020243_

Round 1

Reviewer 1 Report

Comments and Suggestions for Authors

In this paper, the authors investigated the interaction (and co-localization) of Xenopus Sox11 with several partners important for neurogenesis. The results are clear and interesting. But there are some issues that need to be addressed:

1) The authors concluded that Xenopus Sox11 does not interact with Neurog1. This is different from what has been know about their counterparts in mice. To firmly establish this, it is important to use the murine proteins are the positive controls. To make it even better, it would be interesting to determine sequence difference that contribute this outcome.

2) The title is a bit over-claiming: the term “land scape” is for investigating many partners systematically. I would suggest to tune it down a bit.

3) Fig 3 legend: An extra space in the second word of the following phrase “protein-protei n interactions” .

Comments on the Quality of English Language

It is OK.

Author Response

Thank you for all of your thoughtful suggestions. I hope that our changes have enhanced the manuscript and satisfy your requests.  

reviewer 1

In this paper, the authors investigated the interaction (and co-localization) of Xenopus Sox11 with several partners important for neurogenesis. The results are clear and interesting. But there are some issues that need to be addressed:

1) The authors concluded that Xenopus Sox11 does not interact with Neurog1. This is different from what has been know about their counterparts in mice. To firmly establish this, it is important to use the murine proteins are the positive controls. To make it even better, it would be interesting to determine sequence difference that contribute this outcome.

Additional text has been added (lines 189 to 202) to discuss  potential sequence differences that are related to our data. We are working on another study that will examine the requirement of specific amino acids.

I believe you are also suggesting that we test mouse Sox11 with mouse proteins in our in vitro assay. This is a good experiment but we currently do not have the version of mouse SOX11 or mouse NGN1 that can be in vitro transcribed and translated. For your information, we do know that mouse SOX11 interacts with mouse NGN1 in Hek293 cells whereas Xenopus Sox11 doesn’t not interact with mouse NGN1 in these cells. This is an incomplete experiment however and would require additional constructs to be made in order to test other interactions. Unfortunately, the group studying Mouse cortical development has left Georgetown and research science making it difficult to perform these experiments.

2) The title is a bit over-claiming: the term “land scape” is for investigating many partners systematically. I would suggest to tune it down a bit.

We agree and thank you for this suggestion. We have changed the name to “Xenopus Sox11 Partner Proteins and Functional Domains”

3) Fig 3 legend: An extra space in the second word of the following phrase “protein-protei n interactions”.

This has been corrected.

Reviewer 2 Report

Comments and Suggestions for Authors

In this manuscript, Singleton et al. studied the SoxC family transcription factor Sox11 during neural development in Xenopus, with a specific focus on Sox11 interaction with its partner factors. They showed that sox11 is co-expressed with pou3f2 and neurog2 in neural plate. Through in vitro Co-IP experiments, the authors showed that Sox11 interacts with Pou3f2 and Neurog2, but not Neurog1. Furthermore, the study identified that both N-terminal domain and the HMG domain of Sox11 are required for interaction with Pou3f2 and Neurog2.

Sox factors are widely recognized for their interactions with various partners for their specific functions, with many examples previously reported across many contexts. This study focused on Sox11 in Xenopus, and expanded the Sox-partner code during normal development. The manuscript is well written, with clear data presentation and the conclusions are well supported by their analyses.

However, I have a few minor suggestions that could potentially enhance the manuscript:

1.The Co-IP experiments were performed with in vitro translated tagged proteins, would it be possible to perform the Co-IP with embryo lysate, at least for the full-length proteins? The authors used in vitro system, is it due to antibody availability for Sox11, Pou3f2 and Neurog2? If the choice of the in vitro system is due to antibody availability for Sox11, Pou3f2, and Neurog2, this rationale could be explicitly mentioned.

2.Please include information on the sample size for the CoIP/western blot experiment in the method section (e.g. for those in Figure 2).

3. Ideally, some statistical testing needs to be performed for Figure 3H and I, and the methods used for data analysis should also be reported in the method section or figure legends.

Author Response

Thank you for all of your thoughtful suggestions. I hope that our changes have enhanced the manuscript and satisfy your requests.  

Reviewer 2 

In this manuscript, Singleton et al. studied the SoxC family transcription factor Sox11 during neural development in Xenopus, with a specific focus on Sox11 interaction with its partner factors. They showed that sox11 is co-expressed with pou3f2 and neurog2 in neural plate. Through in vitro Co-IP experiments, the authors showed that Sox11 interacts with Pou3f2 and Neurog2, but not Neurog1. Furthermore, the study identified that both N-terminal domain and the HMG domain of Sox11 are required for interaction with Pou3f2 and Neurog2.

Sox factors are widely recognized for their interactions with various partners for their specific functions, with many examples previously reported across many contexts. This study focused on Sox11 in Xenopus, and expanded the Sox-partner code during normal development. The manuscript is well written, with clear data presentation and the conclusions are well supported by their analyses.

However, I have a few minor suggestions that could potentially enhance the manuscript:

1.The Co-IP experiments were performed with in vitro translated tagged proteins, would it be possible to perform the Co-IP with embryo lysate, at least for the full-length proteins? The authors used in vitro system, is it due to antibody availability for Sox11, Pou3f2 and Neurog2? If the choice of the in vitro system is due to antibody availability for Sox11, Pou3f2, and Neurog2, this rationale could be explicitly mentioned.

This is indeed why the in vitro system was used. We have noted that reliable antibodies are not available in lines 193 and 196. 

2.Please include information on the sample size for the CoIP/western blot experiment in the method section (e.g. for those in Figure 2).

Since these are in vitro assays, there is no sample size but only the amount of protein used, which is in the methods section line 124. If you mean replicates, all experiments have been repeated at least 3 times. 

  1. Ideally, some statistical testing needs to be performed for Figure 3H and I, and the methods used for data analysis should also be reported in the method section or figure legends.

A reference to the method used to determine relative band instensity is provided on line 136  in the methods. 

The graph is there to simply represent the relative levels of the 3 replicates. We are not making any claims about the actual amount of reduction in protein interaction. We have changed some of the wording to ensure that we have not misrepresented the data (lines 217-219). We can also remove the graph if the reviewer likes as it is not common practice for co-IP data. 

Reviewer 3 Report

Comments and Suggestions for Authors

This is a very interesting study on the neural development role of Sox11 in Xenopus. The authors discovered that Sox11 interacts with Pou3f2 and Neurog2 and mediates the early neural differentiation process. They further mapped the interaction domains and showed its functional importance. Overall, this study provides valuable information on the role of Sox11 in Xenopus neural development.

I have a few concerns:

Figure 1A. Could the authors provide higher resolution images and scale them up to like Figure 4? The authors also should include the neurog1 is they want to describe the overlap with Sox11 in the text. Also, the asterisk is missing.

Figure 3 title typo: ‘protein-protei n interactions.’

Figure 3A. It would be easier for readers to better understand the domains and their sizes if the authors could add the amino acid numbers at the junction of each domain.

Figure 3 B-G: the authors could make the sequence of HA/FLAG, MYC/FLAG images consistent throughout all panels. They also need to check if the molecular weight label is correct. Why Pou3f2 is below 20 kda in B but below 50 in C and D?

Figure 3 H,I. How is each band normalized? If the bands are not on the same membrane and within similar exposure range, they cannot be compared.

Page 8, line 281. Figure B-C is not about ‘reduced neuron formation’. Did the author cite the wrong figure?

Author Response

Figure 1A. Could the authors provide higher resolution images and scale them up to like Figure 4? The authors also should include the neurog1 is they want to describe the overlap with Sox11 in the text. Also, the asterisk is missing.

We have attempted to provide higher resolution images in the template and will include tif files for publication.

Neurog1 and neurog2  in situ hybridization are extremely similar  and have been published before with images on the online Xenopus database ( https://www.xenbase.org/xenbase/).  The best representation of neurog 1 and neurog2 expression are  in Figure S6 in Riddiford N, Schlosser G. Six1 and Eya1 both promote and arrest neuronal differentiation by activating multiple Notch pathway genes. Dev Biol. 2017 Nov 15;431(2):152-167. However their expression is identical in the neural plate. 

We have made it clearer that we are replicating already published data and the reason we are showing this data with neurog2 as a representative for both neurogenin genes (lines 148-156). 

Figure 3 title typo: ‘protein-protei n interactions.’

This has been corrected

Figure 3A. It would be easier for readers to better understand the domains and their sizes if the authors could add the amino acid numbers at the junction of each domain.

The amino acid numbers have been added in Figure 3A. 

Figure 3 B-G: the authors could make the sequence of HA/FLAG, MYC/FLAG images consistent throughout all panels. They also need to check if the molecular weight label is correct. Why Pou3f2 is below 20 kda in B but below 50 in C and D?

We have checked out molecular weight labels. Pou3f2 is shown below 50kd in 3B, C and D. The size of the Sox11 deletion constructs varies. Delta c-term is below 20kd but delta N-term and delta HMG are below 50kD. 

Figure 3 H,I. How is each band normalized? If the bands are not on the same membrane and within similar exposure range, they cannot be compared.

The proteins that were pulled down by any one antibody ( HA for example) and then probed with any one antibody (e.g Flag) are on the same blot. In other words, in B,C and D, pulldown for HA (Pou3f2) and probed for Flag (sox11 deletion constructs) are on the same blot. Only those samples are compared. 

Page 8, line 281. Figure B-C is not about ‘reduced neuron formation’. Did the author cite the wrong figure?

We did indeed cite the wrong figure and have corrected that